# An enhanced spatial-temporal graph convolution network with high order features for skeleton-based action recognition

**Mohammed H. Al-Hakimi**[1,2]*, **Ibrar Ahmed**[1], **Muhammad Haseeb**[1], **Taha H. Rassem Senior Member IEEE**[3], **Fahmi H. Quradaa**[1,4], **Rashad S. Almoqbily**[1,4]

1 Department of Computer Science, University of Peshawar, Peshawar, Pakistan, 2 Department of Computer Science, Hodeida University, Hodeida, Yemen, 3 School of Computer Science and Informatics, De Montfort University, Leicester, United Kingdom, 4 Department of Computer Science, Aden Community College, Aden, Yemen

☉ These authors contributed equally to this work.

* Hakimi@uop.edu.pk

**Data availability statement:** This work utilized the NTU-RGB+D dataset available at "https://rose1.ntu.edu.sg/dataset/actionRecognition".

## Abstract

Skeleton-based action recognition has emerged as a promising field within computer vision, offering structured representations of human motion. While existing Graph Convolutional Network (GCN)-based approaches primarily rely on raw 3D joint coordinates, these representations fail to capture higher-order spatial and temporal dependencies critical for distinguishing fine-grained actions. In this study, we introduce novel geometric features for joints, bones, and motion streams, including multi-level spatial normalization, higher-order temporal derivatives, and bone-structure encoding through lengths, angles, and anatomical distances. These enriched features explicitly model kinematic and structural relationships, enabling the capture of subtle motion dynamics and discriminative patterns. Building on this, we propose two architectures: (i) an Enhanced Multi-Stream AGCN (EMS-AGCN) that integrates joint, bone, and motion features via a weighted fusion at the final layer, and (ii) a Multi-Branch AGCN (MB-AGCN) where features are processed in independent branches and fused adaptively at an early layer. Comprehensive experiments on the NTU-RGB+D 60 benchmark demonstrate the effectiveness of our approach: EMS-AGCN achieves 96.2% accuracy and MB-AGCN attains 95.5%, both surpassing state-of-the-art methods. These findings confirm that incorporating higher-order geometric features alongside adaptive fusion mechanisms substantially improves skeleton-based action recognition.

## 1 Introduction

Human Action Recognition (HAR) constitutes a pivotal endeavor within the field of computer vision, concentrating on the automatic identification, classification, and prediction of human actions from video data [1]. This undertaking has considerable implications across

**Funding:** The author(s) received no specific funding for this work.

**Competing interests:** The authors have declared that no competing interests exist.

various sectors, including healthcare, sports, surveillance, and human-computer interaction [2–5]. Despite its extensive applicability, HAR continues to pose significant challenges due to the inherent complexity associated with human actions, characterized by variations in motion patterns, alterations in viewpoint, and the occurrence of subtle or difficult-to-detect actions [6,7]. These complexities underscore the necessity for the development of robust and precise recognition systems capable of accommodating diverse and dynamic scenarios.

In recent years, deep learning methodologies have been extensively investigated to improve the efficacy of HAR models. Convolutional Neural Networks (CNNs) have been widely embraced for their ability to capture spatial features in video frames, while Recurrent Neural Networks (RNNs) have been utilized to model temporal dependencies within action sequences [8]. However, these methodologies are predominantly tailored for processing regular grid-like data, such as images or videos, and may encounter difficulties in capturing the structured nature of human motion. In contrast, graph-based approaches, particularly Graph Convolutional Networks (GCNs), have gained traction due to their capacity to model the non-Euclidean structure of skeletal data. By representing human joints as nodes and their physical interconnections as edges, GCNs effectively encapsulate spatial dependencies among skeletal joints. When integrated with temporal information, GCNs can delineate the dynamic evolution of actions, rendering them a formidable tool for skeleton-based HAR [9–12]. Unlike CNNs and RNNs, which process data in a grid-like or sequential format, GCNs leverage the inherent structure of skeletal data, facilitating the modeling of intricate spatial and temporal relationships between joints [9]. This capability has resulted in enhanced performance in action recognition tasks, particularly for actions characterized by complex motion patterns [13].

Despite their achievements, existing GCN-based methodologies also encounter several limitations. First, contemporary models predominantly depend on basic features, such as the xyz-coordinates of joints or bone lengths, which are inadequate for capturing the intricacies of hard-to-detect actions [11]. These fundamental features fail to encapsulate higher-order interactions or subtle motion patterns, thereby constraining the model's capacity to recognize actions necessitating fine-grained analysis [11–14]. While GCNs excel at capturing relationships between proximate joints, they frequently neglect interactions between non-adjacent or distant joints, which are essential for the recognition of complex actions [15]. Some approaches attempt to address these challenges by introducing multi-stream architectures, which process multiple streams of disparate modalities concurrently. However, the computational cost and training complexity grow progressively as more streams are added to the architecture [16,17].

To tackle these limitations, we propose a novel approach that leverages low-level features to create high-order features and introduces a multi-branch Graph Convolutional Network (GCN) architecture for human action recognition. The proposed methodology extends traditional first-order features by capturing correlations between body parts and representing joints and bones from multiple perspectives to identify higher-order interactions and subtle motion patterns. This enhancement is particularly beneficial for recognizing complex actions, especially in the context of fine-grained hand movements and interactions between non-adjacent joints. Additionally, our multi-branch GCN framework processes spatial, temporal, and structural information in parallel, with each branch focusing on different aspects of action data, including local joint interactions, global motion patterns, and temporal dynamics. By fusing the outputs of these branches, a comprehensive representation of the action is generated, enabling the model to learn complementary features and improve adaptability to human action variability. This approach not only addresses the limitations of existing GCN-based methods but also establishes a new benchmark for skeleton-based human action

recognition, particularly in recognizing challenging actions and achieving state-of-the-art performance. The primary contributions of this work are as follows:

1. Novel High-Order Features: We propose high-order features that capture correlations between body parts and represent joints from multiple perspectives. These features enable the model to detect complex and challenging-to-recognize actions by modeling higher-order interactions.

2. Enhanced Multi-Stream Adaptive Graph Convolutional Network (EMS-AGCN): To the best of our knowledge, we are the first to introduce a three-stream network that utilizes high-order representation of joints, bones, and their motions. This multi-modal approach maximizes the use of high-order features to improve recognition performance in demanding scenarios.

3. Early Fusion of Multi-Branch AGCN: We present the early fusion of a multi-branch AGCN framework, integrating joint and bone representations at an early stage. This enhances the model's ability to learn complementary features and improves recognition accuracy.

4. State-of-the-Art Performance with Multi-Stream and Multi-Branch Models: Our proposed models: a multi-stream model that integrates three modalities (joints, bones, and motion) and a multi-branch model that utilizes joint and bone modalities. Both achieve state-of-the-art results and significantly outperform existing methods.

The remainder of this paper is organized as follows: Sect 2 reviews related work in human action recognition, focusing on deep learning and graph-based approaches. Sect 3 outlines the proposed methodology, covering high-order features, multi-stream GCN, and multi-branch GCN architectures. Sect 4 presents the experimental results and analysis, followed by a discussion of their implications in Sect 5. Sect 6 addresses threats to validity and limitations, while Sect 7 concludes with key findings and highlights future research directions.

## 2 Related works

Skeleton-based action recognition has undergone significant evolution over the years, transitioning from traditional methods that rely on handcrafted features to advanced deep learning approaches. Early methodologies primarily concentrated on manually designing features to represent the human body, often employing shallow architectures and domain-specific knowledge [18]. However, these approaches were constrained by limitations, such as the loss of information regarding interactions among body parts and an over-reliance on complex feature engineering [8]. With the rapid evolution of deep learning methodologies and their demonstrated effectiveness across diverse computer vision applications [19], researchers have increasingly adopted deep learning techniques to automatically learn hierarchical representations directly from raw skeleton data, thereby achieving state-of-the-art results. In this work, we categorize these methods into three groups based on the manner in which skeleton data is represented: CNN-based, RNN-based, and GCN-based approaches.

### 2.1 CNN-based approaches

CNN-based methods process skeleton data by converting it into pseudo-images through handcrafted encoding rules, followed by standard CNN classification [20,21]. While these approaches exploit the powerful feature extraction capabilities of CNNs, they often result in a significant loss of critical structural information during the conversion of skeleton data

into grid-like inputs. This limitation severely undermines their capacity to effectively capture the essential spatial relationships between joints, which are crucial for accurate action recognition.

## 2.2 RNN-based approaches

RNN-based methodologies treat skeleton data as a sequence of joint coordinates and employ RNNs to model temporal dependencies. Compared to CNN-based methods, RNN-based approaches are better equipped to capture temporal dynamics [22,23]. However, they are susceptible to issues such as gradient explosion, training difficulties, and considerable computational overhead [24,25]. Although RNN-based methods achieve higher accuracy than traditional approaches, they continue to struggle with effectively modeling the spatial structure of skeleton data [26,27].

## 2.3 GCN-based approaches

To address the shortcomings of CNN- and RNN-based methods, researchers have turned to Graph Convolutional Networks (GCNs), which naturally represent skeleton data as graphs. Yan et al. [9] introduced the Spatial-Temporal Graph Convolutional Network (ST-GCN), a pioneering GCN-based method that models the human skeleton as a spatial-temporal graph. In ST-GCN, each joint corresponds to a graph node, while edges represent both spatial connections (between physically connected joints) and temporal connections (between the same joint across consecutive frames). This approach effectively captures both spatial and temporal dependencies, rendering it a powerful framework for skeleton-based action recognition.

To further enhance ST-GCN, researchers have explored various strategies to model connections between disjoint nodes. For instance, Shi et al. [10] introduced a trainable adjacency matrix to complement the handcrafted graph structure proposed by Yan et al. [9]. Other approaches extend the adjacency graph by calculating node similarity or hop-based distances [28]. Additionally, Obinata et al. [29] proposed new temporal edges that connect a joint to multiple adjacent joints across frames, as well as static spatial edges between the gravity center and all other joints. Despite these advancements, effectively capturing action-based relationships between disjoint nodes remains a challenging issue.

**2.3.1 Multi-stream ST-GCN.** Recent research is focused on enriching skeleton-based action recognition by incorporating multiple streams of features derived from raw joint positions, such as motion, speed, and bone information. This approach, initially proposed by Shi et al. [10], who fused joint and bone streams at the final layer. Since then, the field has progressed to include additional streams, such as motion and speed, leading to increasingly complex architectures, including two-stream [10,30], three-stream [31–34], five-stream [31], and even six-stream [13,14]. Fusing streams at the decision layer remains a common strategy, as it facilitates the integration of diverse features for improved accuracy. For a comprehensive overview, refer to Table 1.

Fusing streams at the decision layer presents significant challenges. While it enhances feature representation, it also increases model complexity and computational overhead. Training separate models for each stream complicates the training process and limits the feasibility of end-to-end training [15]. This trade-off between accuracy and efficiency underscores the need for more streamlined approaches to multi-stream fusion. To strike a balance between accuracy and efficiency, our EMS-AGCN model adopts a streamlined approach—processing only three high-order feature streams (joints, bones, and motions) to mitigate complexity while preserving discriminative power

**Table 1**. **Summary of multi-stream techniques utilized in human action recognition.**

| Ref | Input-Stream | Approach |
|---|---|---|
| Shi et al. [10] | Joint and bone streams | Two-stream based on 2s-AGCN |
| Li et al. [30] | Joint and edge streams | Two parallel streams |
| Qi et al. [6] | Joint and bone streams | Two-stream based on original 2s-AGCN incorporated with Spatial temporal attention on residual unit and two kernel for TCN |
| Liuet al. [31] | Relative position, bone and motion streams | Three-stream based on 2s-AGCN,utilizing first-order features |
| Xie et al. [34] | Joint position, joint motion, bone length, bone motion | Four-stream of multi-scale GCN |
| Donget al. [11] | Joint, bone, velocity, acceleration , and Euclidian distance of 3D joints streams | Five-stream, Data augmentation is achieved by rotating skeletal joints within a ±2-degree |
| Li Fet al. [14] | Joint, bone, joint motion, bone motion, joint relative position, And bone relative position streams | Six-stream, The ST-GCN block consists of spatial GCLs and densely connected multiple temporal layers |
| Lee et al. [35] | Joint and bone stream | Hierarchical decomposition of the skeleton graph into multiple sub-graphs, starting from a designated root joint, followed by a six-way ensemble |
| Jang et al. [36] | Joint and bone stream | Two-stream, four-way ensemble; multi-scale GCN and TCN for enhanced feature learning |
| Mehmood et al. [13] | Joint, bone, joint motion, bone motion, joint distance, bone length streams | Six-stream framework using Extended 2s-AGCN with spatial-temporal attention |

**2.3.2 Multi-branch ST-GCN.** To overcome the limitations associated with decision-layer fusion, Song et al. [15] proposed an early fusion strategy that integrates multiple branches into the main ST-GCN branch. In this framework, each branch processes different types of input data, such as joint coordinates, bone lengths, and motion. By fusing these inputs early in the network, the model can capture richer interactions between spatial and temporal features, leading to more robust representations of human motion [16]. Building on this foundation, several studies have further advanced the multi-branch ST-GCN framework, as depicted in Table 2. One study proposed a multi-branch ST-GCN with joint and motion streams, incorporating multi-scale temporal convolutional networks and part attention mechanisms to enhance feature aggregation [17]. Similarly, Yin et al. [37] extended the multi-branch approach by introducing the ST-Joint attention module after each ST-GCN block, which dynamically highlights the most discriminative joints both at the frame level and across the entire temporal sequence. Additionally, Nan et al. [16] explored residual graph convolutional networks (ResGCNs) on individual branches, with the main branch utilizing a 1s-AGCN architecture. However, these methods might fall into suboptimality because they rely on simple fusion techniques, such as direct concatenation or summation, which overlook the varying importance of feature streams (joint, bone, motion) and assume equal contributions from all inputs. To address this issue, we introduce a novel high-order features and an adaptive fusing method to improve the feature representation at later layers.

## 3 Methodology

The proposed methodology, delineated in this section, is systematically divided into two primary steps: a data preprocessing phase and the implementation of a spatial-temporal Graph Convolutional Network (GCN) model for action recognition. The system's input comprises

**Table 2**. **Summary of multi-branch techniques for human action recognition.**

| | Input-Branch | Approach |
|---|---|---|
| Feng et al. [17] | Joint and motion stream | Multi-Scale TCN, Part Attention, Bottleneck, Human Graph Decrease After Every two Blocks of ST-GCN. Multi-hop GCN, [15] is the baseline. |
| Songet al. [15] | Joint Position, bone and motion stream | Multi-hop GCN, Residual GCN Bottleneck blocks of ST-GCN, Part Attention. |
| Yinet al. [37] | Joint position, bone and motion stream | Gait Recognition, 1s AGCN is the baseline incorporated with spatial temporal Attention layer follow the TCN layer. |
| Nan et al. [16] | Joint position, bone and motion stream | New features, ResGCN [15] on Branches, the main branch is 1s AGCN |

skeletal data, represented by three-dimensional coordinates corresponding to 25 joints, as captured by the Kinect sensor (as illustrated in Fig 1). During the preprocessing phase, the raw skeletal data undergoes processing to extract three distinct data streams: joint data, bone data, and motion data. These data streams are subsequently input into a spatial-temporal neural model, which facilitates the generation of the final representation of actions and their classification.

## 3.1 Data preprocessing

The data preprocessing phase is conducted to eliminate noise and extract pertinent geometric features from two fundamental perspectives: one pertaining to the spatial dimension and the other to the temporal component. We initiate this process with the raw data, transforming it into smoothed and normalized coordinates, thereby establishing it as our baseline. To enhance performance, we employ an early fusion strategy utilizing a multi-branch approach, which incorporates nine channels for each joint, bone, and motion stream. The initial three features for each branch are derived from the formulas presented by the baseline [10].

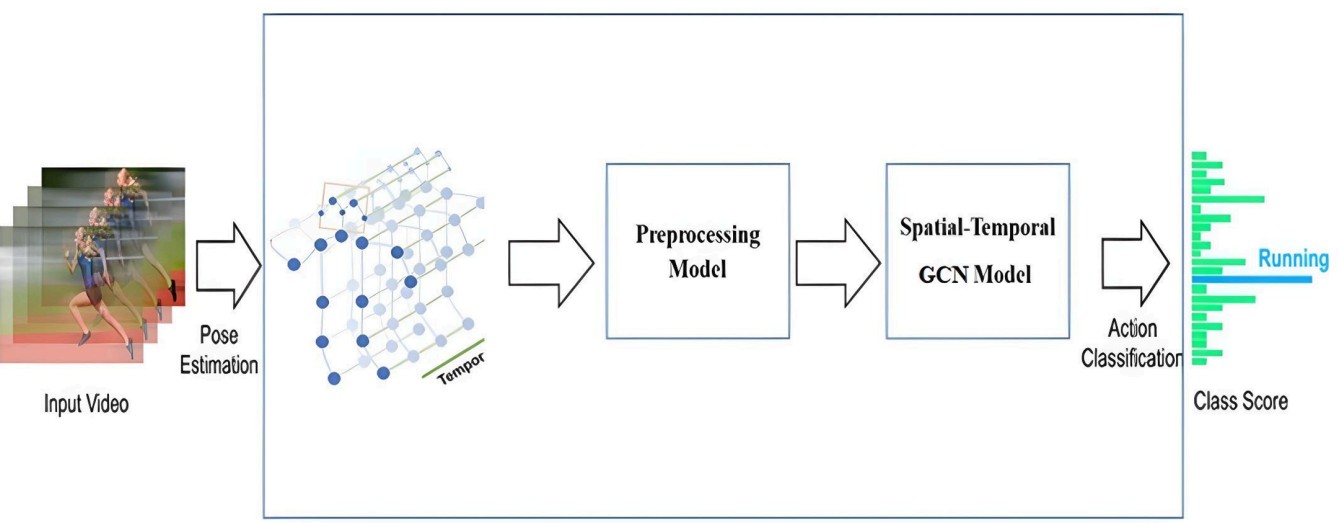

**Fig 1. The proposed methodology.**

Our principal contribution is the introduction of additional geometric features that emphasize the importance of spatial and temporal dynamics.

**3.1.1 Joint-branch.** Let $R^m$ denotes the average coordinates of all joints for person $m$ over all frames, and $P^{j,t}$ denote the coordinates of the central joint of the specific part to which the joint $j$ of frame $t$ belongs (e.g., Neck or Hip). This branch describes the coordinates of the 25 joints, with each joint undergoing three types of normalization as follows:

$$V_j = \left( X_j^t - X_c^t, Y_j^t - Y_c^t, Z_j^t - Z_c^t, \right.$$
$$X_j^t - R_x^m, Y_j^t - R_y^m, Z_j^t - R_z^m,$$
$$\left. X_j^t - P_x^{j,t}, Y_j^t - P_y^{j,t}, Z_j^t - P_z^{j,t} \right) \tag{1}$$

where:

- $(X_j^t, Y_j^t, Z_j^t)$ are the coordinates of joint $j$ at frame $t$.
- $(X_c^t, Y_c^t, Z_c^t)$ are the coordinates of the selected central point of the skeleton at frame $t$.
- $(P_x^{j,t}, P_y^{j,t}, P_z^{j,t})$ are the coordinates of the central joint of the specific part to which joint $j$ of frame $t$ belongs.
- $(R_x^m, R_y^m, R_z^m)$ are the average coordinates of all joints for person $m$ over all frames.

$$R_x^m = \frac{\sum_{j=1}^{N} \sum_{t=1}^{T} X_j^t}{N \times T},$$
$$R_y^m = \frac{\sum_{j=1}^{N} \sum_{t=1}^{T} Y_j^t}{N \times T}, \tag{2}$$
$$R_z^m = \frac{\sum_{j=1}^{N} \sum_{t=1}^{T} Z_j^t}{N \times T}$$

To capture joint features, we started with the three coordinates provided by the Kinect sensor $(X_j^t, Y_j^t, Z_j^t)$. Spatial normalization was achieved by subtracting the coordinates of a selected central point of the skeleton $(X_c^t, Y_c^t, Z_c^t)$. Furthermore, we implemented normalization relative to the central point of all frames and to the central joint of each body part in each frame. This approach explicitly encodes the kinematic relationships between joints, part roots, and action centers, enabling the model to discriminate between highly similar action classes (e.g., reading versus writing) through enhanced feature differentiation.

**3.1.2 Motion-branch.** This branch captures motion dynamics by computing differences between joint coordinates at time steps $t$, $t+1$, and $t+2$, as illustrated in Eq 3. It leverages a temporal window of 5 frames centered around the current frame $t$ to estimate second-order derivatives numerically [16]. The motion features $M_j'$ are as follows:

$$M_j' = \left( X_j^{t+1} - X_j^t, Y_j^{t+1} - Y_j^t, Z_j^{t+1} - Z_j^t, \right.$$
$$X_j^{t+2} - X_j^t, Y_j^{t+2} - Y_j^t, Z_j^{t+2} - Z_j^t,$$
$$\left. X_j^{t+2} + X_j^{t+2} - 2 * X_j^t, Y_j^{t+2} + Y_j^{t-2} - 2 * Y_j^t, Z_j^{t+2} + Z_j^{t-2} - 2 * Z_j^t \right) \tag{3}$$

where

- The first row of Eq 3 delineates the first-order derivatives (velocity) between consecutive frames.

- The second row encapsulates the displacement over a two-frame interval.
- The third row calculates the second-order derivatives (acceleration) utilizing a symmetric five-frame window.

Integrating information regarding speed and acceleration augments the temporal representation of actions, thereby facilitating a more profound understanding of their progression over time. By incorporating these temporal dynamics, the neural network acquires the capability to more effectively discern the nature and intensity of human actions. This enhanced representation significantly improves the model's ability to detect subtle variations, achieving more reliable and accurate activity recognition.

**3.1.3 Bone-branch.** The bone-branch framework seeks to capture structural information by delineating bone properties through four distinct feature types: position, length, angle, and Euclidean distances to spinal joints (defined as central points of body segments, including the neck and hip) and a designated reference point. The characteristics of the bone $B'_j$ connecting joints $u$ and $v$ are articulated as follows.

$$
\begin{aligned}
B'_j = \big( & X^t_u - X^t_v, Y^t_u - Y^t_v, Z^t_u - Z^t_v, \\
& (X^t_u + X^t_v)/2, (Y^t_u + Y^t_v)/2, (Z^t_u + Z^t_v)/2, \\
& \Theta_x, \Theta_y, \Theta_z, \\
& D_N, D_R, D_H \big)
\end{aligned}
\tag{4}
$$

where

$$
D_b = \sqrt{(X^t_u - X^t_v)^2 - (Y^t_u - Y^t_v)^2 + (Z^t_u - Z^t_v)^2}
$$

$$
\theta_x = \arccos\left(\frac{X^t_u - X^t_v}{D_b}\right), \quad \theta_y = \arccos\left(\frac{Y^t_u - Y^t_v}{D_b}\right)
$$

$$
\theta_z = \arccos\left(\frac{Z^t_u - Z^t_v}{D_b}\right)
$$

$$
D_N = \sqrt{(X^t_u - X^t_{\text{neck}})^2 - (Y^t_u - Y^t_{\text{neck}})^2 + (Z^t_u - Z^t_{\text{neck}})^2}
$$

$$
D_R = \sqrt{(X^t_u - R^m_x)^2 - (Y^t_u - R^m_y)^2 + (Z^t_u - R^m_z)^2}
$$

$$
D_H = \sqrt{(X^t_u - X^t_{\text{Hip}})^2 - (Y^t_u - Y^t_{\text{Hip}})^2 + (Z^t_u - Z^t_{\text{Hip}})^2}
$$

The integration of bone segment lengths, angles, and Euclidean distances encodes the structural characteristics of human motion within the neural network. By integrating bone length data, the network can interpret body segment scaling and relative proportions, enhancing its capacity to recognize actions across diverse body configurations. Furthermore, bone angles relative to the axes encode key details about joint flexion, extension, and body posture. By incorporating this anatomical viewpoint, the network better interprets actions through skeletal dynamics, allowing for a richer and more precise analysis of human activity. Collectively, these features empower the network to capture the biomechanical and structural intricacies of human motion, thereby augmenting its effectiveness in action recognition tasks.

## 4 Enhanced multi-stream AGCN model

To incorporate the proposed features, we have developed the Enhanced Multi-Stream AGCN architecture, as depicted in Fig 2. This model integrates three feature streams—joints, bones,

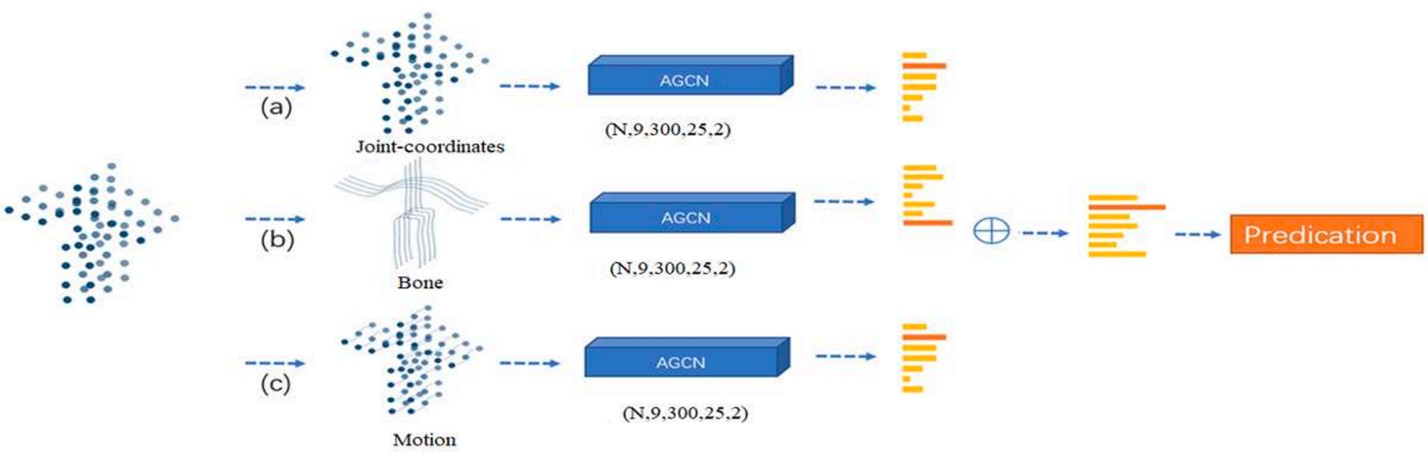

**Fig 2. Architecture of the enhanced multi-stream adaptive graph convolutional network (EMS-AGCN).**

and motion—into the baseline 2s-AGCN framework. Each stream is processed independently through the baseline architecture, utilizing a batch normalization layer at the input and a global average pooling layer at the output to ensure consistent feature map dimensions. Inspired by [11], the final scores from all streams are aggregated through a weighted summation, where joints, bones, and motion are assigned higher weights due to their fundamental significance, whereas velocity and acceleration features, which enhance temporal relationships, are assigned lower weights. The combined score is computed as follows:

$$\lambda = \lambda_{joint} W_{joint} + \lambda_{bone} W_{bone} + \lambda_{motion} W_{motion} \tag{5}$$

where $\lambda_{joint}$, $\lambda_{bone}$, and $\lambda_{motion}$ represent the scores associated with joint, bone, and motion, respectively. $\lambda$ denotes the final score, while $W$ signifies the weights assigned to these scores.

## 5 Multi-branch AGCN model

The proposed methodology for the multi-branch Adaptive Graph Convolutional Network (MB-AGCN) is depicted in Fig 3. The joint and bone streams are channeled into two distinct AGCN branches, which are subsequently fused adaptively into the primary branch. The primary branch comprises six sequential blocks of the proposed spatial-temporal model (elaborated in Sect 3.3.2), which integrates an Adaptive Graph Convolutional Network (AGCN) and a Temporal Convolutional Network (TCN) characterized by a kernel size of $9 \times 1$. Within the primary branch, the Channel Attention Residual (CAR) mechanism is incorporated as the residual unit. The numerical annotations above the blocks denote the input channels, output channels, and stride, respectively. This architecture culminates in a global average pooling layer followed by a dense layer. This approach not only diversifies the input features but also optimizes computational efficiency, thereby facilitating the proposed architecture's capacity to manage larger volumes of input data while attaining improved performance.

### 5.1 Fusion strategy

In order to integrate multiple modalities—specifically joint, bone, and motion data—into the primary branch of a spatial-temporal model, we implement three distinct fusion strategies:

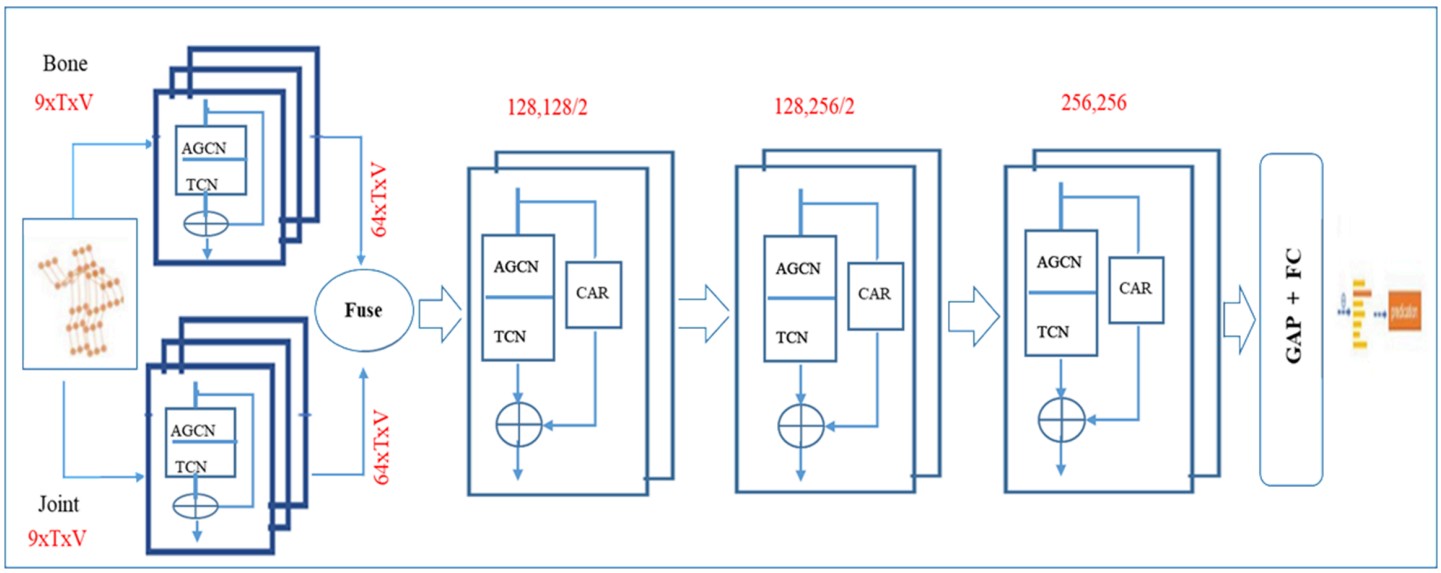

**Fig 3. Architecture of the multi-branch adaptive graph convolution network (MB-AGCN).**

adaptive fusion, simple concatenation, and concatenation with Squeeze-and-Excitation (SE). These strategies are incorporated within the fusion component, as depicted in Fig 4.

**5.1.1 Adaptive fusion.** As illustrated in Fig 4, each incoming branch, consisting of 64 channels, is independently processed through a Squeeze-and-Excitation (SE) layer prior to fusion [38]. The SE layer enhances feature representations by adaptively weighting channels based on their learned interdependencies. The mechanism helps the model focus on important features and downplay less useful ones, which strengthens the representation of each modality prior to integration.

**5.1.2 Simple concatenation.** This method involves the direct concatenation of features derived from the input branches along the channel dimension, without any additional processing. This approach preserves the inherent relationships among modalities, resulting in computational efficiency; however, it may be constrained in its capacity to model complex interactions.

**5.1.3 Concatenation with SE.** In this methodology, features from all modalities are first concatenated along the channel dimension. The concatenated features are then processed through a shared Squeeze-and-Excitation (SE) layer to model the channel interdependencies. The output from the SE layer is subsequently integrated into the primary branch of the spatial-temporal model. While this approach seeks to enhance feature representation through channel-wise attention, it may pose challenges in preserving modality-specific relationships.

## 5.2 Spatial–temporal adaptive graph convolutional module

The Spatial–Temporal Adaptive Graph Convolutional Module is conceptualized based on the Adaptive Graph Convolutional Networks described in the baseline paper [10]. As illustrated in Fig 5(a), the module integrates both spatial and temporal graph convolutions. Each convolution is followed by a batch normalization layer and a ReLU activation function. Additionally, a channel-attention residual connection is incorporated into each block to enhance gradient flow and improve training stability. The spatial graph convolution operation is

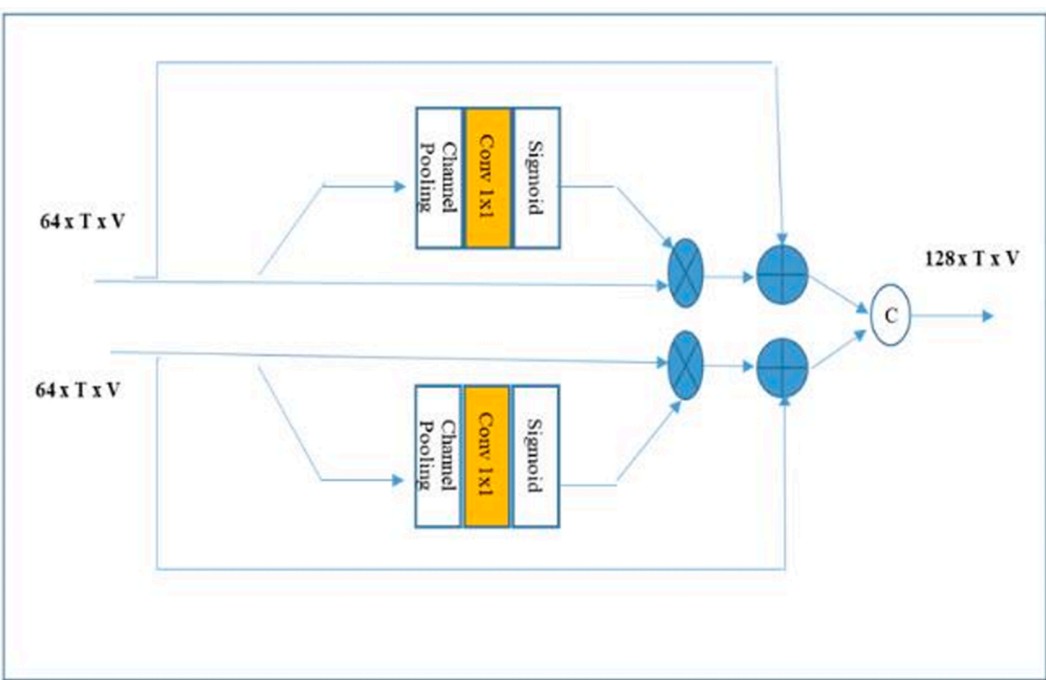

**Fig 4. Architecture of the proposed adaptive fusion model.**

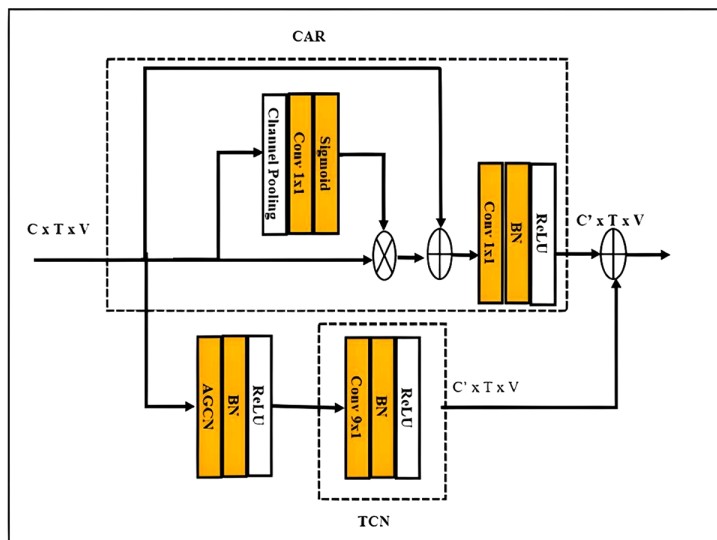

(a) Spatio-temporal block *architecture*

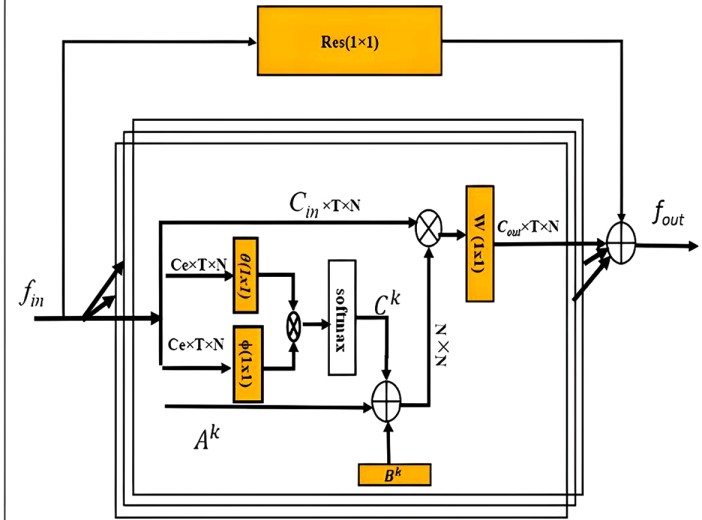

(b) *AGCN architecture [7].*

**Fig 5. (a) Illustrates the structure of the spatio-temporal block within the main branch, whereas (b) Provides a detailed description of the adaptive graph convolutional layer.**

defined as follows:

$$f^h = \sum_{k=0}^{K} W^k f^{(h-1)} (A^k + B^k + C^k) \tag{6}$$

where $f^{(h-1)}$, $f^h$ represent the input and output of layer $h$ respectively. Also, $A^k$, $B^k$ and $C^k$ are the learnable graph and data-based graph respectively, while $K$ represents the number of subsets.

The implementation of this operation is depicted in Fig 5(b). The learnable parameters, emphasized within an orange box, underscore their significance in the model's adaptability. The data-based graph is constructed by measuring the similarity between two vertices in an embedding space using the dot product operation, as illustrated in Eq 7.

$$C^k = softmax(f_{in}^T W_{\Theta k}^T W_{\Phi k} f_{in}) \tag{7}$$

Given the input feature map $f_{in}$ of size $C_{in} \times T \times N$, it is embedded into $C_e \times T \times N$ using two embedding functions, $\theta$ and $\phi$, which are implemented as $1 \times 1$ convolutional layers. The embedded feature maps are then reshaped into an $N \times C_e \times T$ matrix and a $C_e \times T \times N$ matrix. These matrices are multiplied to produce an $N \times N$ similarity matrix, where each element represents the similarity between vertices $v_i$ and $v_j$. The values of the matrix are normalized to the range ($[0,1]$), thereby establishing soft edges between the vertices.

As illustrated in the Fig 5, the layer integrates three distinct graph types: $A^k + B^k + C^k$. The orange box emphasizes the learnable parameters. The notation ($1 \times 1$) specifies the kernel size of the convolution, while K represents the number of subsets involved. The symbol $\oplus$ denotes elementwise summation, whereas $\otimes$ signifies matrix multiplication. The residual connection, indicated by a dotted line, is necessary only when the number of input channels $C_{in}$ differs from the number of output channels $C_{out}$.

## 6 Experiments

Our experiments assess the proposed models using the NTU-RGB+D 60 dataset, examining the effects of various input branches (joints, bones, motion), multi-stream integration, and fusion strategies. We compare our findings with state-of-the-art methods, demonstrating that our EMS-AGCN and MB-AGCN models significantly enhance recognition accuracy and robustness for complex actions through their innovative high-order features and adaptive fusion techniques.

### 6.1 Datasets

To evaluate the efficacy of the proposed methodology, we utilized the NTU-RGB+D dataset [26], which encompasses 56,000 video clips across 60 action categories, performed by 40 subjects aged between 10 and 35 years. Each action is recorded using three Kinect cameras positioned at angles of –45°, 0°, and 45°, thereby providing a range of perspectives while maintaining a consistent camera height. The dataset includes 3$D$ joint localizations for each frame, with each skeleton sequence comprising 25 joints per subject and a maximum of two subjects per video. In alignment with the methodology presented in [31], we implemented two standard benchmarks: Cross-Subject (CS), which partitions the dataset into 40,320 training clips and 16,560 validation clips featuring distinct subjects, and Cross-View (CV), where the training set consists of 37,920 clips from cameras 2 and 3, while the validation set contains 18,960 clips from camera 1.

### 6.2 Experiments setup

The proposed model was trained for 40 epochs utilizing the Stochastic Gradient Descent (SGD) optimizer with Nesterov momentum to estimate the weights of the neural network.

The SGD optimizer was configured with an initial learning rate of 0.1, a momentum of 0.9, and an initial weight decay of 0.0001. The learning rate was reduced by a factor of 10 at the 25th and 35th epochs, with training concluding at the 40th epoch. To enhance computational efficiency and reduce memory usage, SGD processes data in mini-batches rather than employing the entire dataset in each iteration. To ensure that the regularization strength scales appropriately with the learning rate, thereby promoting more stable and effective training, we introduced adaptive weight decay. This mechanism dynamically adjusts the weight decay parameter during training, doubling it each time the learning rate decreases. A batch size of 32 was employed for training, while testing was conducted with a batch size of 64. The weights of the convolutional layers were initialized using the Kaiming normal distribution, thereby ensuring stable and efficient training from the outset.

## 6.3 Ablation study

In this section, we assess the efficacy of the proposed components within our model utilizing the X-View benchmark on the NTU-RGB+D dataset [26]. The initial performance of the 2s-AGCN [10] on the NTU-RGB+D dataset is reported at 95.1%. Through the integration of a refined learning rate scheduler and specifically designed data preprocessing techniques, the performance is enhanced to 96.2%, establishing this as the baseline for our experiments. Additional details are available in the supplementary material.

**6.3.1 The influence of input branch baseline performance.** In this section, we examine the effect of the extracted features on the baseline architecture. As demonstrated in Table 3, our novel features yield consistent performance enhancements across all streams, resulting in an accuracy increase of 0.9% in the Joints Stream, 1.1% in the Bone Stream, and 0.2% in the Motion Stream, as depicted in Fig 6. The combined Joints & Motion Stream achieved an accuracy of 93.99%, which is slightly lower than the individual performances, suggesting potential challenges in the integration of the two streams. Overall, our features exhibited consistent improvements across all streams, although their effectiveness varied depending on the modality.

**6.3.2 Evaluation of enhanced multi-stream AGCN.** In this subsection, we examine the influence of the extracted features on the proposed architecture of EM-AGCN. As illustrated in Table 4, the two-stream combination of Joint and $Bone_1$ Streams yields an improvement of 0.6% in accuracy. In contrast, the three-stream configuration comprising Joint, $Bone_1$, and Motion Streams exhibits a more pronounced enhancement of 1.1 %, which can be attributed to the significant impact of the newly extracted features. Although the two-stream combination of Joint and Motion Streams is not documented in existing literature, our methodology achieves a 0.2% improvement relative to the 2s AGCN. These findings demonstrate that our features consistently improve accuracy, especially in more complex multi-stream settings

As illustrated in Fig 7, the EMS-AGCN model exhibits a high degree of accuracy, surpassing 95% across 43 classes, including "Take off jacket" (99.68%) and "Hopping" (99.68%).

**Table 3. Comparative analysis of accuracy across various input modalities on the NTU-RGBD 60 dataset.**

|  | Stream | Baseline Features (3 features) | Our Features (9 Features) |
|---|---|---|---|
| 1 | Joints Stream | 93.7 | 94.6 |
| 2 | Motion Stream | 92.4 | 92.6 |
| 3 | $Bone^1$ Stream | 93.2 | 94.3 |
| 4 | Joints & Motion | N/M | 93.99 |

*Note:* $Bone^1$ refers to bone features including Bone Length, Bone Position, and Distance.

## THE INFLUENCE OF TCN OF THE INPUT BRANCH

**Fig 6. Performance impact of the proposed features on the baseline model.**

**Table 4. Comparisons of the accuracy of multi-stream with different input modalities on the NTU-RGB+D 60 dataset.**

|  | Input-Streams | Baseline Features(3 features) | Our Features (9 Features) |
|---|---|---|---|
| 1 | Joints & Motion | N/M | **95.3** |
| 2 | Joint & Bone$_1$ | 95.1 | **95.7** |
| 3 | Bone$_1$ & Motion | N/M | **95.65** |
| 4 | Joints & Bone$_1$ & Motion (Our EMS-AGCN model) | 95.5 | **96.2** |

This exceptional performance can be attributed to the diverse joint, motion, and bone streams, which collectively encapsulate a broad spectrum of distinctive movement patterns. Conversely, certain actions, such as "Writing" (75.00%), "Reading" (78.40%), and "Typing on a keyboard" (79.64%), demonstrate lower accuracy owing to their similar skeletal structures [12]. These actions are represented by only two finger joints (the tips of the hand and thumb), complicating differentiation based solely on skeletal data. They rely on fine-grained details that are less discernible in skeletal data alone. These findings underscore the model's robust generalization capability for actions with distinct motion patterns, while also highlighting its limitations in distinguishing subtle or overlapping movements in the absence of additional visual cues.

**6.3.3 The influence of fusion strategies on multi-branch AGCN architecture.** In this subsection, we examine the impact of various fusion strategies on the proposed model across multiple modalities, assessing their efficacy in conjunction with the Channel Attention Residual (CAR) component. The results presented in Table 5 indicate that the effectiveness of these fusion strategies is significantly influenced by the type of input data. For the Joint & Motion modality, adaptive fusion- which applies channel attention individually prior to concatenation-achieves an accuracy of 94.92%, surpassing that of simple concatenation,

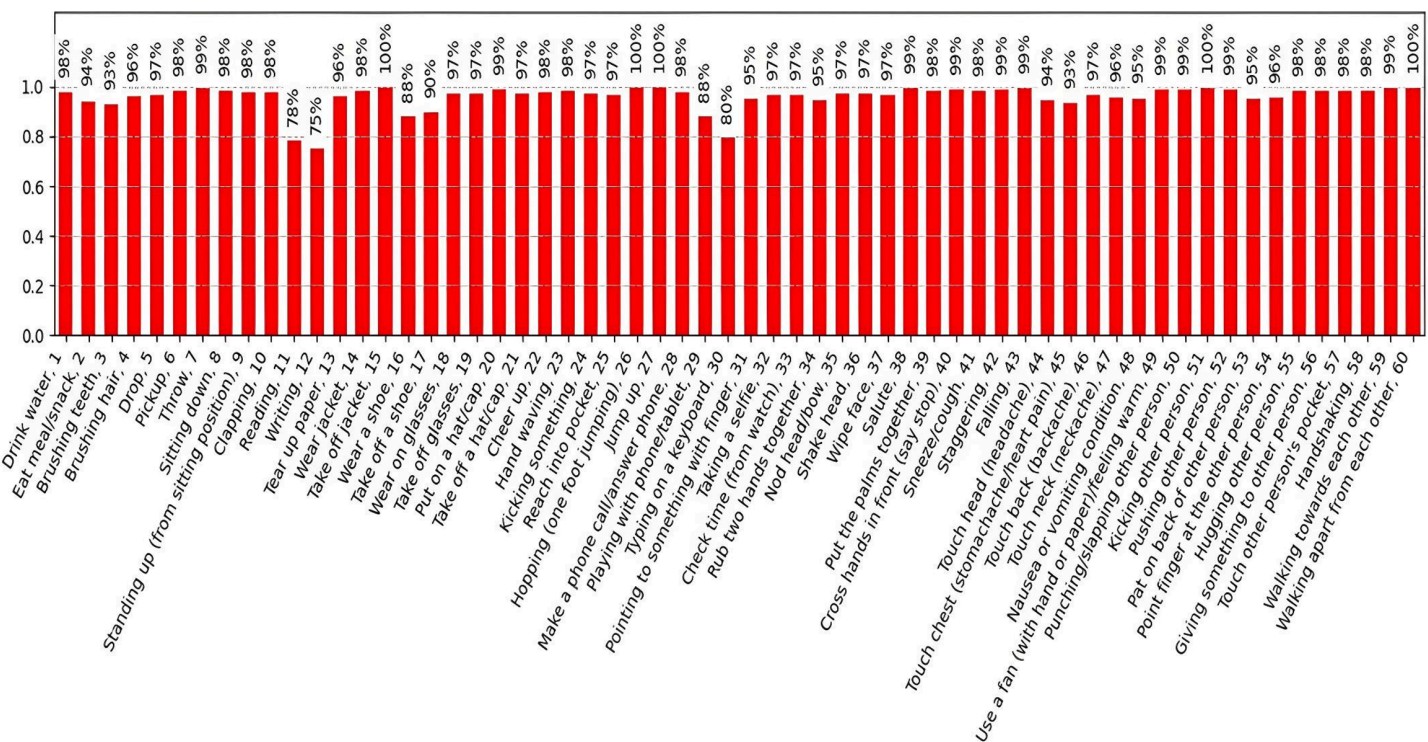

**Fig 7. Cross-view performance evaluation of EMS-AGCN on the NTU-RGB+D 60 dataset.**

**Table 5. Comparisons of the accuracy with different input modalities on the NTU-RGBD 60 dataset.**

| | Input Streams | Fusing Strategy | Residual | Top 1% | Top 5% |
|---|---|---|---|---|---|
| 1 | Joint& Motion | Simple concatenation | Baseline | 94.4 | 99.17 |
| 2 | Joint& Motion | Adaptive fusing | Baseline | **94.92** | 99.22 |
| 3 | Joint& $Bone_1$ | Simple concatenation | Baseline | 94.79 | 99.2 |
| 4 | Joint& $Bone_1$ | Adaptive fusing | Baseline | 94.68 | 99.22 |
| 5 | Joint& $Bone_1$ | Simple concatenation | CAR | 94.86 | 99.17 |
| 6 | Joint& $Bone_2$ | Simple concatenation | CAR | **95.5** | 99.3 |
| 7 | Joint& $Bone_2$ | Simple concatenation + Channel Attention | CAR | 94.9 | 99.2 |
| 8 | Joint& $Bone_2$ | Adaptive fusing | CAR | **95** | 99.3 |

*Note:* $Bone^1$ refers to bone features such as bone length, bone position, and distance, whereas $Bone_2$ refers to features including bone length, angles, and distance.

which yields an accuracy of 94.4%. This finding suggests that adaptive recalibration is particularly advantageous for motion data, where dynamic temporal patterns require nuanced modeling.

In contrast, for the Joint& $Bone_1$ modality, characterized by a high correlation between bone positions and joint relative positions, simple concatenation outperforms adaptive fusion (94.79% vs. 94.68%). In this context, the additional complexity introduced by channel attention residuals leads to overfitting and a decline in performance.

For Joint& Bone$_2$, where bone positions are replaced with bone angles, simple concatenation demonstrates a more effective alignment with the data representation and outperforms adaptive fusion. Furthermore, the integration of channel attention residuals in the main branch further enhances performance, resulting in a peak accuracy of 95.5 %.

These findings underscore the necessity of tailoring fusion strategies to the specific characteristics of the input data, as no single approach achieves optimal performance across all modalities.

**6.3.4 Comparisons with state-of-the-art.** In this subsection, we evaluate the performance of the proposed models and compare them with leading methods in the field. Table 6 presents a comprehensive comparison of recent human action recognition approaches utilizing the NTU-RGB+D 60 dataset. The results demonstrate a discernible trend of increasing accuracy over time, beginning with TS-LSTM (74.6% X-Sub, 2017) [39] and progressing to advanced graph convolution network (GCN)-based and multi-stream architectures, such as EMS-AGCN (96.2% X-View, proposed) and MSTGCN (91.3% X-Sub, 2022) [17].

Multi-stream methods, including 2s-AGCN (88.5% X-Sub, 95.1 X-View) [10] and STMGCN (90.2% X-Sub) [6], consistently exhibit enhanced performance by capitalizing on complementary spatial and temporal features. More lightweight models like MSTGCN and STI-GCN [40] underscore the significance of achieving a balance between efficiency and accuracy. However, the enduring disparity between cross-subject (X-Sub) and cross-view (X-View) accuracy highlights the challenges associated with generalization across subjects.

As illustrated in Fig 8, both the MB-AGCN and EMS-AGCN models demonstrate commendable performance in recognizing actions characterized by distinct, large-scale motions, achieving nearly equivalent accuracy in instances such as "Take off jacket" (MB-AGCN: 100.00%, EMS-AGCN: 99.68%). However, EMS-AGCN exhibits superior performance in fine-grained actions, exemplified by "Brushing hair" (EMS-AGCN: 96.26% vs. MB-AGCN: 95.65%) and "Clapping" (EMS-AGCN: 97.68% vs. MB-AGCN: 92.35%). This advantage can be attributed to its integration of motion features, which more effectively capture temporal dynamics and subtle movements. Nevertheless, both models encounter challenges with actions necessitating exceptionally fine-grained detail, such as "Writing" (MB-AGCN: 72.99%, EMS-AGCN: 75.00%), thereby underscoring the necessity for additional modalities, such as appearance or object context, to enhance performance further.

Focusing specifically on MB-AGCN, as depicted in Fig 9, the proposed MB-AGCN model exhibits robust performance across the majority of action categories. A significant number of actions, including "Take off jacket" (100%) and "Falling" (99.37%), achieve accuracies

**Table 6**. Comparisons of the test results with state-of-the-art methods on the NTU-RGB+D 60 dataset.

| Method | Model Size (M) | Approach | Year | Accuracy (%) | |
|---|---|---|---|---|---|
| | | | | X-Sub | X-View |
| TS-LSTM [39] | – | – | 2017 | 74.6 | 81.3 |
| ST-GCN [9] | 3.1 | One-Stream | 2018 | 81.5 | 88.3 |
| AS-AGCN [41] | 6.99 | One-Stream | 2019 | 86.8 | 94.2 |
| MSGCN [42] | – | Two-Stream | 2020 | 88.8 | 95.7 |
| 2s-AGCN [10] | 6.94 | Two-Stream | 2019 | 88.5 | 95.1 |
| STI-GCN [40] | 1.6 | – | 2020 | 90.1 | **96.1** |
| MSTGCN [17] | 0.94 | Two-Branch | 2022 | 91.3 | 95.9 |
| STMGCN [6] | – | Multi-Stream | 2023 | 90.2 | **95.5** |
| Spatio-temp. [14] | 3.43 | Three-Branch | 2024 | 89.53 | **95.34** |
| Ours EMS-AGCN | 6.94 | Multi-Stream | – | – | **96.2** |
| Ours MB-AGCN | 3.6 | Multi-Branch | – | 90.4 | **95.5** |

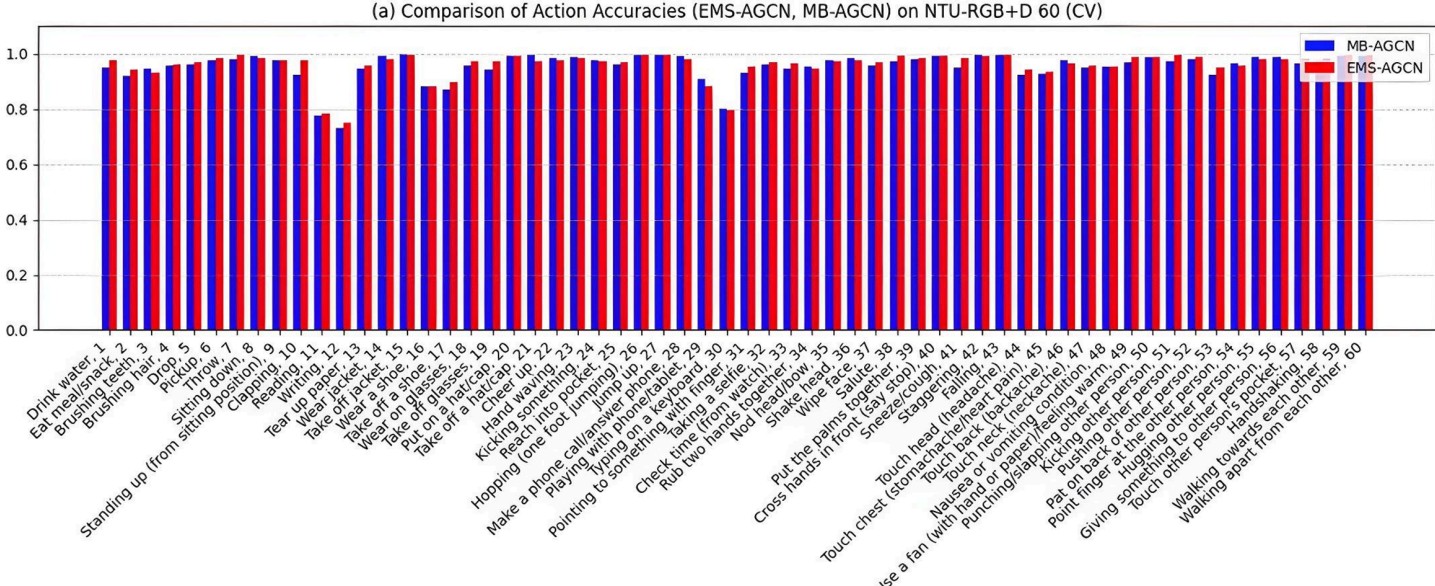

**Fig 8. Comparison of EMS-AGCN and MB-AGCN.**

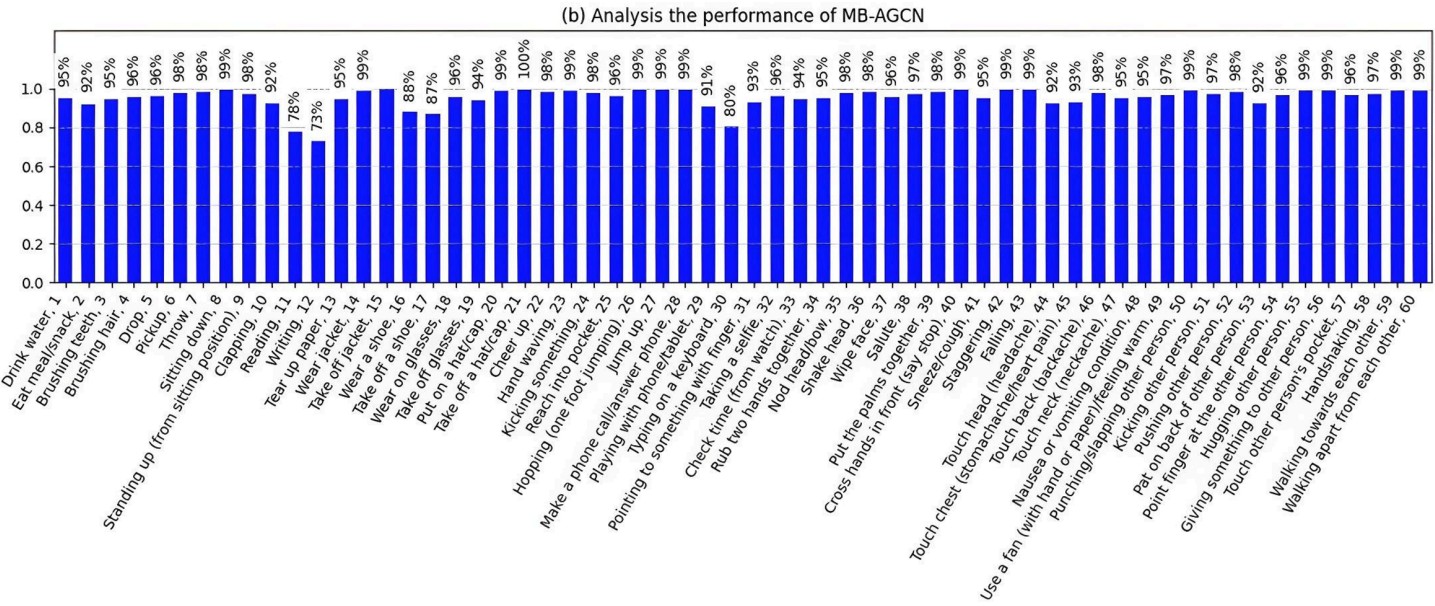

**Fig 9. Accuracy of MB-AGCN using joint and bone modalities.**

exceeding 90%, with several approaching perfect accuracy. Even fine-grained actions such as "Writing" (72.99%) and "Typing on a keyboard" (80.36%) surpass 70% accuracy, representing a notable improvement over prior studies, in which such actions typically yielded accuracies as low as 60% [14,35]. This enhancement is attributed to the model's feature extraction framework, which effectively captures spatial-temporal dependencies within skeletal data, facilitating robust modelling of both global and localized motion patterns.

The model exhibits an exceptionally high accuracy rate (exceeding 99%) for tasks involving distinct, large-scale motions, including actions such as "Take off jacket" and "Hopping (one foot jumping)" (99.37%). These actions are characterized by repetitive, full-body movements that are readily identifiable due to their pronounced spatial-temporal patterns. Likewise, actions that involve object interaction, such as "Take off a hat/cap" (99.68%), or social interaction, such as "Hugging another person" (98.71%), benefit from additional contextual cues, which further enhance recognition accuracy. This indicates that the model performs remarkably well in contexts defined by significant, large-scale motions or enriched with contextual information.

Conversely, the model encounters challenges when tasked with fine-grained actions that depend on subtle, localized movements, such as "Writing" (72.99%) and "Wearing a shoe" (88.04%). These actions do not exhibit distinct spatial-temporal patterns and demonstrate considerable intra-class variability. For example, the action of "Writing" can vary significantly among individuals, influenced by differences in posture, writing style, and surface interaction. Similarly, actions such as "Drinking water" (94.94%) and "Brushing teeth" (94.74%) pose difficulties due to their variable execution across different individuals and contexts.

As demonstrated in the confusion matrix presented in Fig 10, the model struggles to differentiate between similar actions, including "Reading," "Writing," and "Typing on a keyboard," which are frequently misclassified. This misclassification results from the shared skeletal patterns of these actions and their reliance on fine-grained hand and finger movements that are challenging to distinguish using skeleton data alone. This highlights the necessity for more advanced methodologies, such as the incorporation of appearance or object context, to address intra-class variability and improve performance for fine-grained and underrepresented actions.

## 7 Discussion

The experimental results substantiate the efficacy of the proposed approach, demonstrating that enhanced input features, multi-stream architectures, and optimized fusion strategies collaboratively yield significant performance improvements in action recognition. The comprehensive evaluation reveals consistent enhancements across all tested configurations, affirming the robustness of our methodology. Notably, the extracted features contributed to increased accuracy in the Joints and $Bone_1$ streams, while multi-stream configurations and adaptive fusion strategies further augmented overall performance. Experimental results show that our EMS-STGCN model sets a new performance benchmark on the NTU-RGB+D 60 dataset, particularly excelling in cross-view evaluation compared to current state-of-the-art methods.

### 7.1 The influence of the input branch

The enhancement observed in the Joints and Bone1 streams illustrates the efficacy of our extracted features in capturing spatial and structural relationships. Conversely, the lack of improvement in the Motion stream indicates that motion data may require alternative feature representations or regularization techniques to mitigate overfitting. This underscores the necessity of customizing feature extraction methods to align with the distinct characteristics of each modality.

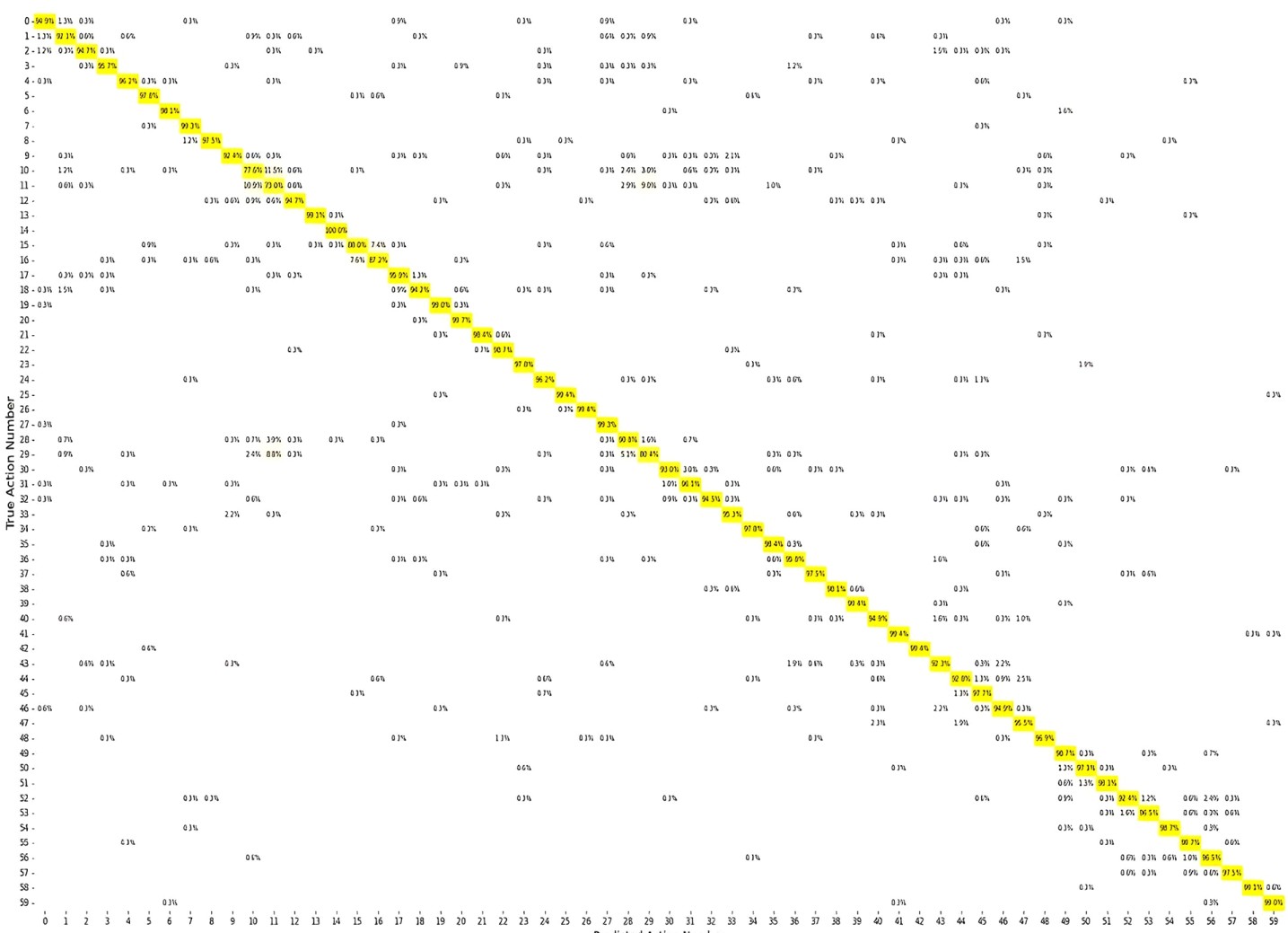

**Fig 10. The confusion matrix of the MB-AGCN model applied to the cross-view protocol of the NTU-RGB+D 60 dataset.**

## 7.2 Multi-streams architecture

The three-stream configuration yielded the most substantial improvement in accuracy, demonstrating the advantages of leveraging complementary information from multiple modalities. However, this enhancement is accompanied by increased computational complexity, thus highlighting a trade-off between performance and efficiency. Future research may investigate lightweight architectures that sustain performance while minimizing computational overhead.

## 7.3 Influence of the fusion strategy

These findings elucidate key trade-offs inherent in fusion strategies. Adaptive fusing performs exceptionally well with joint and motion data due to its capacity to dynamically recalibrate feature responses, yet it encounters challenges with bone data, likely due to incompatible feature representations. While simple concatenation is effective for joint and bone data, it exhibits poor performance on motion data, where adaptive modelling is crucial.

The concatenation with SE strategy demonstrates suboptimal performance across all modalities, suggesting that the application of SE post-concatenation disrupts inter-modal relationships. In conclusion, these results emphasize the necessity for modality-specific fusion strategies within multi-branch action recognition frameworks. No singular approach consistently excels across all modalities, highlighting the importance of aligning fusion techniques with the intrinsic characteristics of the input data.

### 7.4 Comparisons with state-of-the-art methods

The proposed model, EMS-AGCN, achieves state-of-the-art performance on the NTU-RGB+D 60 dataset, attaining a cross-view accuracy of 96.2%, thereby surpassing existing methodologies. The results indicate a discernible trend of increasing accuracy over time, with EMS-AGCN and MB-AGCN capitalizing on complementary spatial and temporal features to achieve superior performance. Furthermore, while the model excels in large-scale motions and object-interaction actions, it encounters difficulties with fine-grained actions such as "Writing," attributable to subtle motions and intra-class variability.

## 8 Threats to validity and limitations

This study encounters two primary challenges, as illustrated in Figs 8, 9 and 10: fine-grained action recognition and intra-class variability in action recognition. Although our model demonstrates proficiency in recognizing fine-grained actions, such as "Typing on a keyboard," its performance within these categories is comparatively lower than that for broader action classes. This discrepancy highlights the inherent difficulty in capturing subtle movements and fine motor skills, which tend to be less distinctive than larger motions, such as "Hopping." To address this issue, advanced techniques, including hierarchical attention mechanisms, could be considered to enhance the isolation of fine-grained features.

## 9 Conclusion and future work

In this study, we present a novel methodology for human action recognition that integrates enhanced input features, multi-stream configurations, and early fusion of multi-branch strategies. The experimental results indicate substantial improvements over the baseline model, with the proposed features leading to increased accuracy in the Joints and Bone streams. Adaptive fusion strategies have demonstrated effectiveness for motion and joint data, while the three-stream configuration has capitalized on complementary spatial and temporal information to further enhance performance. Furthermore, the Channel Attention Residual unit in MB-AGCN has augmented critical spatio-temporal information by emphasizing key nodes in similar actions, thereby improving recognition accuracy.

The proposed methodology has been evaluated on the NTU-RGB+D 60 dataset, where EMS-AGCN achieved an accuracy of 96.2% and MB-AGCN achieved 95.5%, both surpassing the baseline 2s-AGCN and other models, particularly in the recognition of similar human actions. Despite these advancements, challenges persist in cross-subject generalization and the modelling of fine-grained actions.

Building upon the contributions of this work, our future research will focus on integrating the proposed features within a multi-scale Graph Convolution Network (GCN) architecture. This architecture is ideal for hierarchically modeling the human skeleton, capturing fine-grained motions and complex limb interactions simultaneously. We expect this multi-level approach to significantly improve recognition of actions involving both local and global

dynamics. Such advancements could enhance applications in areas like human-computer interaction, automated sports analytics, and clinical rehabilitation monitoring.

## Author contributions

**Conceptualization:** Mohammed H. Al-Hakimi, Ibrar Ahmed, Taha H. Rassem.

**Data curation:** Mohammed H. Al-Hakimi.

**Formal analysis:** Mohammed H. Al-Hakimi, Ibrar Ahmed, Muhammad Haseeb, Fahmi H. Quradaa, Rashad S. Almoqbily.

**Investigation:** Mohammed H. Al-Hakimi, Taha H. Rassem.

**Methodology:** Mohammed H. Al-Hakimi, Ibrar Ahmed, Muhammad Haseeb, Taha H. Rassem, Fahmi H. Quradaa, Rashad S. Almoqbily.

**Project administration:** Mohammed H. Al-Hakimi, Muhammad Haseeb, Taha H. Rassem, Fahmi H. Quradaa.

**Resources:** Mohammed H. Al-Hakimi, Muhammad Haseeb, Taha H. Rassem.

**Software:** Mohammed H. Al-Hakimi, Ibrar Ahmed, Muhammad Haseeb, Taha H. Rassem, Rashad S. Almoqbily.

**Supervision:** Mohammed H. Al-Hakimi, Ibrar Ahmed, Muhammad Haseeb, Rashad S. Almoqbily.

**Validation:** Mohammed H. Al-Hakimi, Ibrar Ahmed, Taha H. Rassem, Fahmi H. Quradaa, Rashad S. Almoqbily.

**Visualization:** Mohammed H. Al-Hakimi, Ibrar Ahmed, Fahmi H. Quradaa.

**Writing – original draft:** Mohammed H. Al-Hakimi, Fahmi H. Quradaa, Rashad S. Almoqbily.

**Writing – review & editing:** Mohammed H. Al-Hakimi, Fahmi H. Quradaa, Rashad S. Almoqbily.

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
