## [Decision Letter · Decision Letter 0]

15 Aug 2025

PONE-D-25-35630An Enhanced Spatial-Temporal Graph Convolution Network with High Order Features for Skeleton-Based Action RecognitionPLOS ONE

Dear Dr. Al-Hakimi,

Thank you for submitting your manuscript to PLOS ONE. After careful consideration, we feel that it has merit but does not fully meet PLOS ONE’s publication criteria as it currently stands. Therefore, we invite you to submit a revised version of the manuscript that addresses the points raised during the review process.

We look forward to receiving your revised manuscript.

Kind regards,

Guangyin Jin

Academic Editor

PLOS ONE

3. Please note that PLOS One has specific guidelines on code sharing for submissions in which author-generated code underpins the findings in the manuscript. In these cases, we expect all author-generated code to be made available without restrictions upon publication of the work. Please review our guidelines at https://journals.plos.org/plosone/s/materials-and-software-sharing#loc-sharing-code and ensure that your code is shared in a way that follows best practice and facilitates reproducibility and reuse.

4. PLOS requires an ORCID iD for the corresponding author in Editorial Manager on papers submitted after December 6th, 2016. Please ensure that you have an ORCID iD and that it is validated in Editorial Manager. To do this, go to ‘Update my Information’ (in the upper left-hand corner of the main menu), and click on the Fetch/Validate link next to the ORCID field. This will take you to the ORCID site and allow you to create a new iD or authenticate a pre-existing iD in Editorial 

Additional Editor Comments (if provided):

Reviewers' comments:

Reviewer's Responses to Questions

**Comments to the Author**

1. Is the manuscript technically sound, and do the data support the conclusions?

Reviewer #1: Yes

2. Has the statistical analysis been performed appropriately and rigorously? 

Reviewer #1: Yes

3. Have the authors made all data underlying the findings in their manuscript fully available?

Reviewer #1: Yes

4. Is the manuscript presented in an intelligible fashion and written in standard English?

Reviewer #1: Yes

5. Review Comments to the Author

Reviewer #1: 1- Summarize the abstract so that it covers the proposed algorithm in a simplified manner.

2- There is no need to refer to reference [9] twice in this paragraph, but to refer to it only once in the second location: The data preprocessing phase is conducted to eliminate noise and extract pertinent geometric features from two fundamental perspectives: one pertaining to the spatial dimension and the other to the temporal component. We initiate this process with the raw data, transforming it into smoothed and normalized coordinates [9], thereby establishing it as our baseline. To enhance performance, we employ an early fusion strategy utilizing a multi-branch approach, which incorporates nine channels for each joint, bone, and motion stream. The initial three features for each branch are derived from the formulas presented by the baseline [9]. Our principal contribution is the introduction of additional geometric features that emphasize the importance of spatial and temporal dynamics.

3- Add references published in the years 2024-2025, with at least two references for each year mentioned above.

4- Showcase future work more broadly and clearly

6. PLOS authors have the option to publish the peer review history of their article (what does this mean?). If published, this will include your full peer review and any attached files.

Reviewer #1: No

---

## [Author Response · Author response to Decision Letter 1]

1 Sep 2025

Reviewer Response Report

I would like to express my sincere gratitude to the Editor and the reviewer. Your valuable feedback and insights are greatly appreciated, and we are committed to addressing all your comments and suggestions to enhance the quality and impact of our work.

Editor's Journal comments and responses

• Comment 1: [Please ensure that your manuscript meets PLOS ONE's style requirements, including those for file naming. The PLOS ONE style templates can be found at….]

Response : We would like to inform you that we have adhered to PLOS ONE's style requirements as outlined in the PLOS ONE style templates.

• Comment 2: [We suggest you thoroughly copyedit your manuscript for language usage, spelling, and grammar. If you do not know anyone who can help you do this, you may wish to consider employing a professional scientific editing service.]

• Response: We sincerely thank the reviewers and the editorial office for their suggestion regarding the language quality of our manuscript. In response, we have carefully and thoroughly revised the manuscript to improve grammar, spelling, and overall readability. These revisions were performed by the authors and colleagues with strong proficiency in academic English. Please refer to the revised version of the manuscript for the updated content..

• Comment 3: [Please note that PLOS ONE has specific guidelines on code sharing for submissions in which author-generated code underpins the findings in the manuscript. In these cases, all author-generated code must be made available without restrictions upon publication of the work.]

Response : Thank you for your concerns regarding the PLOS ONE guidelines on code sharing requirement. We assure you that we are fully committed to transparency and are willing to provide any details about the source code to reviewers upon request.

• Comment 4: [PLOS requires an ORCID iD for the corresponding author in Editorial Manager on papers submitted after December 6th, 2016.]

Response : As per the journal’s requirements, I have created an ORCID iD for the corresponding author (0009-0003-4633-7530), which has also been updated in Editorial Manager.

• Comment 5: [ If the reviewer comments include a recommendation to cite specific previously published works, please review and evaluate these publications to determine whether they are relevant and should be cited. There is no requirement to cite these works unless the editor has indicated otherwise.]

Response : We carefully reviewed the reviewer comments, and we did not find any recommendation to cite specific previously published works.

• Comment 6: [ Please review your reference list to ensure that it is complete and correct. If you have cited papers that have been retracted, please include the rationale for doing so in the manuscript text, or remove these references and replace them with relevant current references. Any changes to the reference list should be mentioned in the rebuttal letter that accompanies your revised manuscript.]

Response : We have carefully reviewed the reference list to ensure that it is complete and accurate. None of the cited works are retracted. Additionally, as suggested, we have incorporated four recent references from 2024 and 2025 to further strengthen the manuscript. Kindly refer to the revised version of the manuscript for these updates.

Comments from Reviewer 1 and Responses

• Comment 1: [ Summarize the abstract so that it covers the proposed algorithm in a simplified manner.]

Response : Thank you for your suggestion. We have revised the abstract to provide a concise and simplified summary of the proposed algorithm, highlighting its key components and contributions. Please refer to the updated abstract in the revised manuscript.

Comment 2: [There is no need to refer to reference [9] twice in this paragraph, but to refer to it only once in the second location: ]

Response : Thank you for pointing this out. We have revised the manuscript to cite reference [10] (previously [9], due to the addition of a new reference) only once in the second location, as suggested. This change can be found in the revised manuscript on page 9, line 205.

Comment 3: [ Add references published in the years 2024-2025, with at least two references for each year mentioned above.]

Response : We thank the reviewer for this valuable suggestion. In accordance with the recommendation, we have added four recent references [1-4] (two from 2024 and two from 2025) to strengthen the related work section and ensure the manuscript reflects the most up-to-date research..

• Comment 4: [Showcase future work more broadly and clearly]

Response : We thank the reviewer for this suggestion. In response, we have expanded the future work. Please refer to revised manuscript on page 25, lines 572–577, for the revised text.

References

[1] Xie, J., Meng, Y., Zhao, Y., Nguyen, A., Yang, X., & Zheng, Y. (2024, March). Dynamic semantic-based spatial graph convolution network for skeleton-based human action recognition. In Proceedings of the AAAI conference on artificial intelligence (Vol. 38, No. 6, pp. 6225-6233)

[2] Jang, S., Lee, H., Kim, W. J., Lee, J., Woo, S., & Lee, S. (2024). Multi-scale structural graph convolutional network for skeleton-based action recognition. IEEE Transactions on Circuits and Systems for Video Technology, 34(8), 7244-7258.

[3] Chen, D., Chen, M., Wu, P., Wu, M., Zhang, T., & Li, C. (2025). Two-stream spatio-temporal GCN-transformer networks for skeleton-based action recognition. Scientific Reports, 15(1), 4982.‏.

[4] Zhang, Y., & Wang, Y. (2025). A comprehensive survey on RGB-D-based human action recognition: algorithms, datasets, and popular applications. EURASIP Journal on Image and Video Processing, 2025(1), 15

---

## [Decision Letter · Decision Letter 1]

4 Sep 2025

An Enhanced Spatial-Temporal Graph Convolution Network with High Order Features for Skeleton-Based Action Recognition

PONE-D-25-35630R1

Dear Dr. Al-Hakimi,

We’re pleased to inform you that your manuscript has been judged scientifically suitable for publication and will be formally accepted for publication once it meets all outstanding technical requirements.

Kind regards,

Guangyin Jin

Academic Editor

PLOS ONE

Additional Editor Comments (optional):

Reviewer #1:

Reviewers' comments:

Reviewer's Responses to Questions

**Comments to the Author**

1. If the authors have adequately addressed your comments raised in a previous round of review and you feel that this manuscript is now acceptable for publication, you may indicate that here to bypass the “Comments to the Author” section, enter your conflict of interest statement in the “Confidential to Editor” section, and submit your "Accept" recommendation.

Reviewer #1: (No Response)

2. Is the manuscript technically sound, and do the data support the conclusions?

Reviewer #1: (No Response)

3. Has the statistical analysis been performed appropriately and rigorously? 

Reviewer #1: (No Response)

4. Have the authors made all data underlying the findings in their manuscript fully available?

Reviewer #1: (No Response)

5. Is the manuscript presented in an intelligible fashion and written in standard English?

Reviewer #1: (No Response)

6. Review Comments to the Author

Reviewer #1: (No Response)

7. PLOS authors have the option to publish the peer review history of their article (what does this mean?). If published, this will include your full peer review and any attached files.

Reviewer #1: No

---

## [Editor Report · Acceptance letter]

PONE-D-25-35630R1

PLOS ONE

Dear Dr. Al-Hakimi,

I'm pleased to inform you that your manuscript has been deemed suitable for publication in PLOS ONE. Congratulations! Your manuscript is now being handed over to our production team.

Kind regards,

on behalf of

Dr. Guangyin Jin

Academic Editor

PLOS ONE